# Evaluation of the Training Session in Elite Paralympic Powerlifting Athletes Based on Biomechanical and Thermal Indicators

**DOI:** 10.3390/sports11080151

**Published:** 2023-08-10

**Authors:** Larissa Christine Vieira Santos, Felipe J. Aidar, Rodrigo Villar, Gianpiero Greco, Jefferson Lima de Santana, Anderson Carlos Marçal, Paulo Francisco de Almeida-Neto, Breno Guilherme de Araújo Tinoco Cabral, Georgian Badicu, Hadi Nobari, Raphael Frabrício de Souza, Walderi Monteiro da Silva Júnior

**Affiliations:** 1Graduate Program of Physical Education, Federal University of Sergipe (UFS), São Cristovão 49100-000, Brazil; larissav.fisio@gmail.com (L.C.V.S.); acmarcal.ufs@gmail.com (A.C.M.); raphaelctba20@hotmail.com (R.F.d.S.); walderim@yahoo.com.br (W.M.d.S.J.); 2Group of Studies and Research of Performance, Sport, Health and Paralympic Sports (GEPEPS), Federal University of Sergipe (UFS), São Cristovão 49100-000, Brazil; 3Department of Physical Education, Federal University of Sergipe (UFS), São Cristovão 49100-000, Brazil; 4Graduate Program of Physiological Science, Federal University of Sergipe (UFS), São Cristovão 49100-000, Brazil; 5Cardiorespiratory & Physiology of Exercise Research Laboratory, Faculty of Kinesiology and Recreation Management, University of Manitoba, Winnipeg, MB R3T 2N2, Canada; rodrigo.villar@umanitoba.ca (R.V.); santanaj@myumanitoba.ca (J.L.d.S.); 6Department of Translational Biomedicine and Neuroscience (DiBraiN), University of Study of Bari, 70124 Bari, Italy; gianpiero.greco@uniba.it; 7Department of Physical Education, Federal University of Rio Grande do Norte (UFRN), Natal 59078-970, Brazil; paulo220911@hotmail.com (P.F.d.A.-N.); brenotcabral@gmail.com (B.G.d.A.T.C.); 8Department of Physical Education and Special Motricity, Transilvania University of Brasov, 500068 Brasov, Romania; georgian.badicu@unitbv.ro; 9Faculty of Sport Sciences, University of Extremadura, 10003 Cáceres, Spain; hadi.nobari1@gmail.com; 10Graduate Program of Health Science, Federal University of Sergipe (UFS), São Cristovão 49100-000, Brazil

**Keywords:** disabled persons, asymmetry, recovery, Paralympic sports

## Abstract

Background: Paralympic powerlifting (PP) is performed on a bench press, aiming to lift as much weight as possible in a single repetition. Purpose: To evaluate thermal asymmetry and dynamic force parameters with 45 and 80% 1 Repetition Maximum (1 RM) in PP athletes. Methods: Twelve elite PP male athletes were evaluated before and after a training session regarding skin temperature (thermography) and dynamic force indicators (Average Propulsive Velocity-MPV, Maximum Velocity-VMax, and Power). The training consisted of five series of five repetitions (5 × 5) with 80% 1 RM. The force indicators and dynamics before and after (45% 1 RM) were evaluated in series “1” and “5” with 80% 1 RM. Results: The temperature did not present asymmetry, and there were differences between the moment before and after. In MPV, Vmax, and Power, with 45% 1 RM, there were differences both in asymmetry and in moments (*p* < 0.005). With 80% 1 RM, asymmetry was observed, but no differences between moments (*p* < 0.005). Conclusion: No thermal asymmetry was observed. There were reductions in MVP and VMax at 45 and 80% 1 RM but without significant differences between time points (before and after). However, there was asymmetry in the moments before and after within a safety standard, where Paralympic powerlifting was safe in terms of asymmetries.

## 1. Introduction

Powerlifting is a strength sport that consists of three lifts: squat, deadlift, and bench press, with the goal of lifting the heaviest weight possible in a single attempt [1,2,3]. Although the rules are similar to those of conventional powerlifting, in Paralympic sports, Paralympic powerlifting is a strength sport that only has the adapted bench press, in which athletes have their legs on the bench not on the floor [4,5]. As in conventional powerlifting, in Paralympic powerlifting, the athlete who lifts the heaviest weight wins the competition [5].

It has been reported that Paralympic athletes exhibit greater asymmetry compared to conventional athletes, particularly in impairments that have unilateral relationships [6,7,8]. However, there are studies reporting that the type of disability, and their possible asymmetries, would not affect the result [9]. On the other hand, the rules of the Paralympic sport, compared to the conventional one, are very strict in relation to the symmetry of the movement, where even small asymmetries during heavy weightlifting can invalidate the attempt to lift, thus requiring an adaptation in terms of greater symmetry in Paralympic athletes [5]. Strength training normally uses larger, high-intensity loads, which would affect movement velocity, promote hormonal changes, and change local temperature [10,11,12,13], leading to thermal changes and increased fatigue [14,15] and a consequent increase in movement asymmetry [16]. Previous studies reported that fatigue would be associated with greater asymmetries in movement and thermals, with a consequent increase in the risk of injuries [16,17,18,19].

Asymmetries tend to interfere with sports performance, and this relationship between asymmetry and worsening performance has been reported in other sports, with an increased risk of injury [20,21,22]. On the other hand, some studies have shown that training can reduce asymmetries [23,24]. Given this context, some challenges arise in relation to Paralympic powerlifting (PP), such as the rules that do not allow asymmetries in movement, the intensity of training, and its consequences on fatigue and increases in asymmetries, whether in movement or thermals. Thus, strength training at a high intensity would induce acute and short-term fatigue [25], affecting dynamic strength parameters (e.g., velocity and power) [26,27]. Fatigue would also affect movement symmetry due to muscle overload and would even be reflected in thermal asymmetries [28,29].

Thus, taking into account that Paralympic athletes would present greater asymmetries, and also considering that fatigue and training could affect these possible asymmetries, the objectives of the present study were to evaluate thermal asymmetry and dynamic force parameters in PP athletes (a) before and after a training session performed at 45% one-maximum repetition (1 RM), and (b) before and after the first and last series of five sets of five repetitions performed at 80% of 1 RM. It is hypothesized that PP athletes would show more thermal asymmetry and altered dynamic force parameters (mean propulsive velocity, maximum velocity, and power) after a training session at 45% and 80% of 1 RM.

## 2. Materials and Methods

### 2.1. Study Design

This study is a crossover design, where each participant acted as their own control. The study was carried out over two weeks. In the first week, participants engaged in a familiarization session and a bench press one-repetition maximum (1 RM) test. In the second week, athletes performed four repetitions at 45% of 1 RM followed by a conventional Paralympic powerlifting training of five sets of five repetitions (5 × 5). After that, athletes repeated the four repetitions at 45% of 1 RM. Linear encoders and surface thermography were used to estimate dynamic force parameters (mean propulsive velocity, maximum velocity, and power) and skin temperature (infrared thermography), respectively. Force parameters and skin temperature data were assessed for the dominant and non-dominant arm [30,31], as shown in Figure 1.

### 2.2. Sample

The sample consisted of 12 male elite PP athletes. As inclusion criteria, all athletes had at least 18 months of competitive experience in the sport and were eligible to compete according to the Brazilian Paralympic Committee (BPC) rules (IPC, 2023). These athletes were ranked among the top 10 in their respective bodyweight categories. Regarding disabilities, four athletes had lower limb malformations (arthrogryposis); four had lower limb amputations; two had spinal cord injuries below the eighth thoracic vertebrae, and two had poliomyelitis sequelae.

The sample power was calculated a priori using the open-source software G*Power^®^ (Version 3.0; Berlin, Germany), choosing an “F-family statistic (ANOVA)” considering a standard α < 0.05, β = 0.80 and the effect size of 1.4 found for Mean Propulsive Velocity (MPV) [32]. Thus, a sample power of 0.80 (very strong) was estimated for a sample of 12 participants. Athletes participated voluntarily in the study and signed an informed consent form, in accordance with Resolution 466/2012 of the National Commission for Ethics in Research (CONEP) of the National Health Council, following the ethical principles expressed in the Helsinki Declaration (1964, reformulated in 2013) of the World Medical Association. This study was approved by the Ethics Committee on Research at the Federal University of Sergipe, CAAE: 2.637.882 (approval date: 7 May 2018). The sample characterization is shown in Table 1.

### 2.3. Instruments

Participants’ body mass was measured on a digital platform scale (Michetti, São Paulo, SP, Brazil), with a maximum capacity of 300 kg and dimensions of 1.50 × 1.50 m. Athletes were assessed while seated, considering their physical disabilities. For the intervention and 1 RM assessments, we used an official bench press bench (210 cm), a barbell (220 cm), and weight plates (Eleiko, Halmstad, Sweden) approved by the International Paralympic Committee [5].

The dynamic force parameters (MPV, Vmax, and Power) were recorded using a Vitruve encoder (Vitruve, Madrid, Spain) [33]. The analysis of these parameters was performed before and after a training session using a load of 45% of 1 RM, where the velocity would be close to 1.0 m.s^−1^ [34,35]. These dynamic parameters were also compared between the first and last series of the five sets of five repetitions (5 × 5).

Infrared thermography was used to measure skin temperature, in which athletes were instructed to remain seated and as relaxed as possible to avoid interference with the measurements. Athletes were also instructed not to perform any physical exercise in the 24 h before testing, as well as to avoid consuming caffeine, stimulants, and alcohol [16]. The tests were performed in a quiet room, with temperature between 22 and 24 °C, and relative humidity of approximately 50%, measured by a Thermo-Hygrometer Hikari HTH-240 (Hikari, Shenzhen, China).

Thermographic images were obtained using a Seek Thermal Compact Pro thermal camera (Seek Thermal, Moscow, Russia). This camera has a resolution of 320 × 240 pixels and operates in a temperature range between −4 and 330 °C at distances between 0.91 m and 5.48 m. The images were collected from the clavicular region of the major pectoralis and long head of the triceps brachii [14,36]. Figure 2 displays the two linear encoders (Figure 2A) and the thermographic images (Figure 2B,C). The linear encoders (Vitruve Force Measurement System, Mostoles, Madrid, Spain) [33,37] accurately measured vertical displacement velocity [38]. These encoders were used to determine maximum velocity (Vmax), mean propulsive velocity (MPV), and power.

### 2.4. Procedures

In the first week, participants were familiarized with the procedures and test protocols followed by the determination of 1 RM. Participants started attempts with a weight that they could lift only once using maximum effort. If the participant performed more than one movement, increments were added until the maximum load was reached in a single movement, not exceeding three to five attempts. If the athlete could not perform a single repetition, 2.4 to 2.5% of the load was subtracted from the previous attempt. Participants rested 3–5 min between each attempt [39,40]. The subjects rested 3–5 min between attempts [38,41]. This test was conducted 72 h prior to the evaluative process that occurred in the second session. Warming up for the 1 RM test was the same as described in week two below.

In the second week, the intervention started with a warm-up consisting of a 10-min warm-up consisting of 20 repetitions of shoulder abduction with dumbbells, shoulder development, and shoulder rotation with dumbbells. Subsequently, a specific warm-up on the bench press was performed with the barbell weight (20 kg) and 10 slow repetitions (3.0 × 1.0 s, eccentric × concentric) and 10 fast repetitions (1.0 × 1.0 s, eccentric × concentric). Afterward, athletes performed a set of four repetitions at 45% 1 RM with the maximum possible velocity before and after the training session [2,42]. Subsequently, a 5 × 5 protocol at 80% of 1 RM was administered to the athletes [14,43]. Pre- and post-data were collected using a linear encoder on all four repetitions at 45% of 1 RM and in the first, fifth, and last 5 × 5 series at 80% of 1 RM [38,41].

### 2.5. Statistics

Descriptive statistics were performed with measures of central tendency, mean ± standard deviation (X ± SD), and a 95% confidence interval (95% CI). The Shapiro–Wilk test determined the normality of the variables. A Two-Way Analysis of Variance (ANOVA) for repeated measures (two factors) was used to detect possible statistically significant differences between sides (dominant and non-dominant) and moments (before and after 45% or 80% 1 RM). The Bonferroni post hoc test was used to identify the statistically significant main effects and interactions. The level of significance adopted was *p* ≤ 0.05. The partial eta square (η2p) was used for determination of the effect size (small effect ≤ 0.05, medium effect 0.05 to 0.25, high effect 0.25 to 0.50, and very high effect > 0.50) following the cut-off points [44,45]. Statistical analyses were performed using the Statistical Package for the Social Sciences (SPSS) version 25.0 (IBM, New York, NY, USA) and Prisma GraphPad version 8.1 (GraphPad Software, San Diego, CA, USA).

## 3. Results

Table 2 depicts the mean and standard deviation and the 95% confidence interval (95% CI) values regarding skin temperature at the clavicular region of the pectoralis major and triceps brachii long head in the dominant and non-dominant arm before and after a training session. It is noteworthy that the collection of the thermographic images was performed before and after the intervention, as shown in Figure 1.

There were statistically significant differences before and after a training session (*p* < 0.05), but no differences between the dominant and non-dominant arm. Figure 3 shows participants’ individual skin temperature responses in the pectoralis major and triceps brachii muscles in the dominant and non-dominant arm before and after a training session.

Table 3 depicts the mean and standard deviation and 95% confidence interval (95% CI) of the mean propulsive velocity (MPV), maximum velocity (Vmax), and power at 45% 1 RM before and after a training session as well as at the first and fifth series at 80% 1 RM.

The individual MPV, Vmax, and power responses of the dominant and non-dominant arm before and after a training session performed at 45% 1 RM are presented in Figure 4.

The individual MPV, Vmax, and power responses of the dominant and non-dominant arm before and after a training session performed at 80% 1 RM before and after the first and last series of the 5 × 5 are presented in Figure 5.

## 4. Discussion

The objectives of this study were to assess thermal asymmetry and dynamic force parameters in Paralympic powerlifting athletes before and after a training session. The training session consisted of exercises performed at 45% of their one-repetition maximum (1 RM) and five sets of five repetitions at 80% of their 1 RM.

The results revealed both thermal and dynamic force parameter asymmetries. In terms of skin temperature measured using infrared thermography, there were variations between the different moments, but no significant differences in terms of asymmetry were observed. However, when considering dynamic force indicators at 45% of 1 RM, there were differences between the moments and evident asymmetry. On the other hand, at 80% of 1 RM, there were no differences between the moments, but there were significant asymmetries, particularly during movements at a higher intensity.

This suggests the presence of asymmetry in these parameters, especially during movements involving greater intensity.

### 4.1. Skin Temperature

Regarding skin temperature, our study found no asymmetry either before or after the training session. However, there was an increase in skin temperature in dominant and non-dominant limbs between before and after the training session. Studies suggested that physical exercise promotes a muscular inflammatory state, which is associated with an increase in local temperature, remaining elevated for ~24–48 h or more, depending on the workout intensity [39,46]. This temperature increase is normal, as long as it is within a pattern of values, where larger differences have a greater impact and a higher risk of injury [16]. However, it has been observed that infrared thermographic images demonstrate high sensitivity in relation to possible physiological changes in the muscle, especially in the 24 h after training [15]. Another study presented results where the muscle temperature would decrease during the first minute (between the beginning and the end of the first exercise series). And yet, at the end of the third series, there was an increase in the local temperature of approximately 8.4% compared to the initial temperature. A difference of 6.6% was also observed in relation to the control. Thus, skin temperature in high-intensity exercise would decrease in the initial phase, and then continuously increase until muscle fatigue [47]. In the same direction, it was observed that the skin temperature behavior would vary according to the type of exercise, intensity, duration, muscle mass, and subcutaneous fat layer. The kinetics of skin temperature were evaluated on the worked musculature and other body segments during and after exercise, according to the type and intensity of exercise. The temperature behavior was observed during exercise, immediately afterward, and up to 48 h after exercise, in different types and intensities of exercise. Skin temperature in active muscles increased during high-intensity anaerobic exercise, slowly decreased after exercise, and increased again in the days after training. Contrary to this, the local temperature decreased during low-intensity aerobic exercise, returning to normal values a few minutes later and showing a slight increase in the following days [10].

In the present study, no significant differences were observed in thermal asymmetry between the dominant and non-dominant arms, either in the pectoralis major or the triceps brachii. Asymmetry, in this context, refers to mechanical imbalances in corresponding body parts, and greater asymmetry is associated with reduced performance and an increased risk of injury, particularly during strength training [48]. Previous reports suggest that higher degrees of asymmetry can be more injurious, with thermal asymmetries ≤0.4 °C considered normal and ≥1.6 °C considered higher risk, necessitating a halt in sports practice [16]. Our study revealed a pre-training asymmetry of 0.16 °C in the pectoralis major, with similar values in the triceps brachii (dominant vs. non-dominant arm). After training, the asymmetry in the pectoralis major increased to 0.5 °C, and in the triceps brachii, it reached 0.33 °C, both still within the acceptable range [16].

Monitoring training based on temperature is crucial, especially for athletes with physical disabilities who may experience overloading in affected body segments, impacting the performance of certain movements. The use of orthoses, prostheses, and wheelchairs may potentiate injuries and hinder the maintenance of body symmetry, a crucial aspect in competitions and training, particularly in Paralympic powerlifting [23,49]. Our data showed that Paralympic powerlifting training with high loads shows temperatures in a safe range (<1.6 °C) despite thermal asymmetries. This result indicates that, at least for pectoralis major and triceps brachii, Paralympic powerlifting was performed within a safe body temperature.

### 4.2. Dynamic Force Parameters

For the MPV at 80% of 1 RM in the first set, the asymmetry between dominant and non-dominant arm was ~0.15 m.s^−1^, decreasing to 0.07 m.s^−1^ in the last set. Maximum velocity (Vmax) at 80% of 1 RM in the first set (2.24 m.s^−1^) reduced to 0.14 m.s^−1^ at the last set. However, there were no statistically significant differences between moments (before and after) for MPV and Vmax. Our results indicate that the asymmetry was maintained both for the evaluation with 45% and with 80% of 1 RM, in the moments before and after. Izquierdo et al. (2006) showed that mean velocity during bench press was reduced between intensities of 65% and 80% of 1 RM, indicating that time under muscular tension is directly related to intensity, and higher loads compromise adaptations and joint functions, impacting movement symmetry [4,38,50]. Previous studies have indicated that a reduction in velocity is directly related to the intensity and action of a movement, being an important indicator of fatigue [34,35].

Our study showed a higher velocity in the non-dominant limb compared to the dominant one. Some studies, which evaluated the velocity in Para powerlifting, observed differences, notably with higher intensities, where the non-dominant side presented higher velocity than the dominant side [23,32]. Our study showed a higher velocity in the non-dominant limb compared to the dominant one. Some studies, which evaluated the velocity in Para powerlifting, observed differences, notably with higher intensities, where the non-dominant side presented higher velocity than the dominant side [23,51,52]. This could be explained by the lesser control that the non-dominant side would have in relation to the dominant side. In this direction, other studies showed asymmetry, mainly with higher intensities, with higher values for the non-dominant side, a fact observed in our study [23,32].

In general, velocity decreases with higher loads due to the slower contraction shortening of skeletal striated muscle and the greater force generated and vice versa. This inverse relationship between force and velocity is a basic physiological principle related to muscle contraction mechanisms [53,54,55,56]. The force–velocity relationship has increasingly been used for training purposes [57,58]. This relationship would also be a good tool for assessing the consequences of fatigue [35,59], allowing for better control of loads and effort during training [60,61], particularly important for Paralympic powerlift thletes. 

Possible asymmetries have been the subject of study, above all, as a way of controlling training [10,10,16], as a way of monitoring fatigue [17,62,63,64], as a tool for reducing the risk of injury [16,63,65], and even as a way of evaluating the performance of athletes [20,22]. In this sense, the monitoring of possible asymmetries tends to assist training, especially high-intensity ones, aimed at gaining strength [21,66]. On the other hand, in Para powerlifting, asymmetries have been the target of invalidating movements and consequently limiting performance [5]. Thus, the control of asymmetries has become increasingly studied.

The applicability of training control through velocity, particularly with higher loads, has been discussed in the literature [4,23,41,67], but it remains controversial. The controversies seem to be regarding the relationship between the actual values over a certain period of time [41]. The main arguments are that sports performance is subject to many temporal changes and the need for greater training variation to promote functional adaptations regarding velocity and force [4,23,41,67].

Our study has limitations, such as the fact that the athletes’ diet and sleep habits were not controlled during the study. However, they were instructed to maintain similar routines regarding diet and sleep during the test period. The research sample was small, including only national- and international-level athletes. Therefore, they probably present less asymmetry compared to non-trained populations due to adaptations caused by training and powerlifting characteristics (e.g., modality rules). The relationship between injury level, specific dysfunctions, and possible asymmetries presented were not assessed due to the reduced sample and the fact that many athletes’ injuries affected only one side of their bodies.

## 5. Conclusions

Based on the results, we can infer that skin temperature increased in both the dominant and non-dominant arms of Paralympic powerlifting athletes before and after training sessions. However, there were no significant thermal asymmetries identified between the dominant and non-dominant arm, in either the pectoralis major or the triceps brachii muscles. The detected asymmetry after training remained within an acceptable safety range (<1.6 °C).

Significant reductions were observed in mean propulsive velocity and maximum velocity during evaluations performed with 45% and 80% of the one-repetition maximum (1 RM), both before or during the first set and after or during the last set. Nevertheless, no significant differences were found between these moments (before and after). Despite these changes, the dynamic strength parameters did not compromise the athletes’ safety, even at higher intensities (80% 1 RM).

Despite the observed asymmetry before and after training, the results still fell within acceptable safety standards, suggesting that Paralympic powerlifting training appears to be safe. Additionally, Para powerlifting training seems to promote greater symmetry in accordance with the sport’s rules. Therefore, even when utilizing higher loads characteristic of maximum strength training, the training was deemed safe, and any asymmetries, be it thermal or related to velocity, did not appear to compromise safety during the training.

## Figures and Tables

**Figure 1 sports-11-00151-f001:**
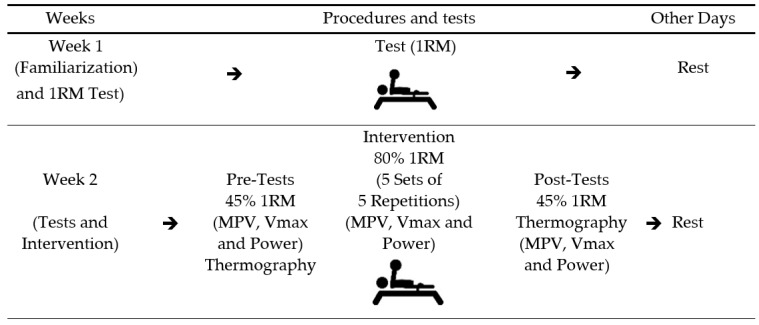
Experimental study design. Legend: 1 RM: one repetition maximum; MVP: mean propulsive velocity; Vmax: maximum velocity.

**Figure 2 sports-11-00151-f002:**
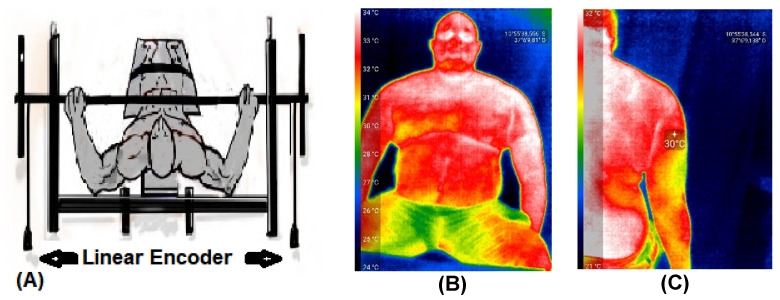
Illustration of the two linear encoders (**A**) and the thermographic images of the pectoralis major (**B**) and triceps brachii (**C**).

**Figure 3 sports-11-00151-f003:**
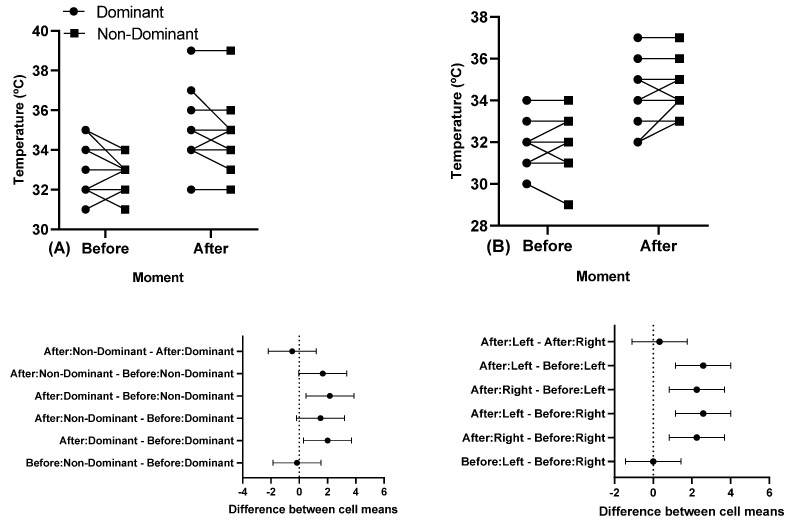
Individual skin temperature responses in the muscles, and below in the same column is the 95% confidence interval of (**A**) the clavicular region of the pectoralis major and (**B**) triceps brachii long head before and after a training session. Note: some athletes presented similar results; thus, all 12 participants’ data are not visible in the (**A**,**B**).

**Figure 4 sports-11-00151-f004:**
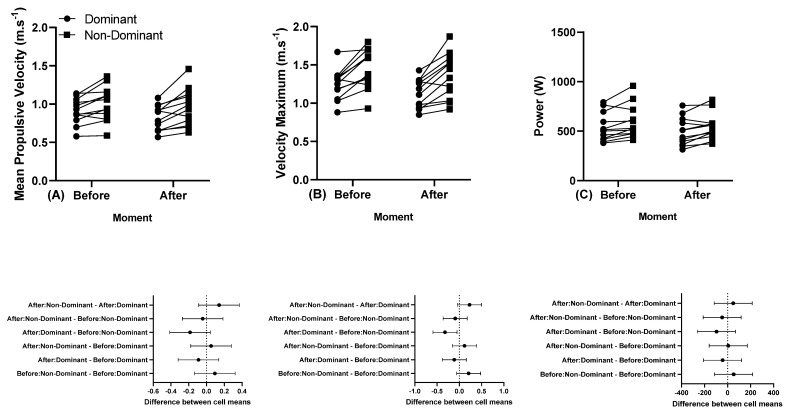
(**A**) Individual mean propulsive velocity, (**B**) individual maximum velocity, and (**C**) individual power, and below in the same column is the 95% confidence interval of the dominant and non-dominant arm before and after a training session performed at 45% 1 RM. Note: some athletes presented similar results; thus, all 12 participants’ data are not visible in the (**A**–**C**).

**Figure 5 sports-11-00151-f005:**
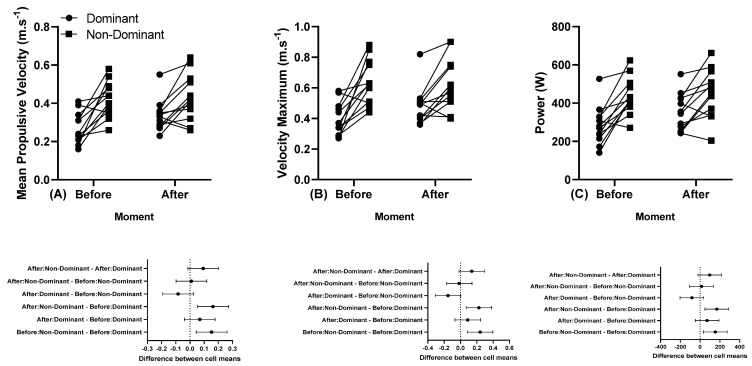
(**A**) Individual mean propulsive velocity, (**B**) individual maximum velocity, and (**C**) individual power, and below in the same column is the 95% confidence interval of the dominant and non-dominant arm before and after the first and last series of the 5 × 5 performed at 80% 1 RM. Note: some athletes presented similar results; thus, all 12 participants’ data are not visible in the (**A**–**C**).

**Table 1 sports-11-00151-t001:** Sample characterization.

Variables	(Mean ± SD)
Sample	12
Age (years)	29.08 ± 6.37
Body mass (kg)	79.17 ± 19.01
Experience (years)	4.42 ± 1.29
1 RM bench press (kg)	146.25 ± 43.80 *
1 RM/body mass	1.87 ± 0.42 **

* The load lift by the athletes ranked them among the top 10 in their categories at the national level. ** 1 RM/body mass values > 1.4 for the bench press is considered elite for athletes (Ball & Wedman, 2018). SD = standard deviation.

**Table 2 sports-11-00151-t002:** Mean ± SD and 95% confidence interval of the skin temperature in the pectoralis major and triceps brachii muscles in the dominant and non-dominant arms before and after a training session.

	Before		After				
	Dominant (a)	Non- Dominant (b)	Dominant (c)	Non- Dominant (d)	*p*-Value	F	η2p
Pectoralis Major (°C)	33.08 ± 1.44 (32.17–34.00)	32.92 ± 1.00 (32.28–33.55)	35.08 ± 1.73 a (33.98–36.18)	34.58 ± 1.73 b (33.78–35.68)	“a” *p* = 0.002 “b” *p* = 0.003	F(1,11) = 18.359	0.625
Triceps Brachii (°C)	31.92 ± 1.08) (31.23–32.61)	31.92 ± 1.31 (31.08–32.75)	34.17 ± 1.47 a (33.23–35.10)	34.50 ± 1.17 b (33.76–35.24)	“a” *p* = 0.001 “b” *p* < 0.001	F(1,11) = 28.641	0.723

Values are mean ± SD and 95% CI of 12 participants. “a”, ”b”: statistically significant differences (*p* < 0.05). η2p = partial eta square (very high effect).

**Table 3 sports-11-00151-t003:** Mean ± SD and 95% confidence interval of the mean propulsive velocity (MPV), maximum velocity (Vmax), and power at 45% 1 RM and first and fifth series at 80% 1 RM in the dominant and non-dominant arm before and after a training session.

	Before		After				
	Dominant (a)	Non-Dominant (b)	Dominant (c)	Non-Dominant (d)	*p*-Value	F	η2p
MPV 45% 1 RM	0.91 ± 0.17 b (0.80–1.02)	1.00 ± 0.22 0.86–1.15)	0.82 ± 0.17 a (0.71–0.92)	0.96 ± 0.24 c (0.81–1.11)	“a” *p* = 0.005 “b” *p* = 0.006 “c” *p* = 0.002	F(1,11) = 8.933 F(1,11) = 16.318	“a” = 0.448 “b,c” = 0.597
Vmax 45% 1 RM	1.24 ± 0.20 b (1.12–1.37)	1.45 ± 0.26 d (1.29–1.62)	1.13 ± 0.19 a (1.01–1.25)	1.36 ± 0.29 c (1.17–1.54)	“a” *p* = 0.011 “b” *p* = 0.001 “c” *p* = 0.001 “d” *p* = 0.009	F(1,11) = 14.987 F(1,11) = 26.214	“a,d” = 0.577 “b,c” = 0.704
Power 45% 1 RM	538.78 ± 142.90 b (447.99–629.57)	591.53 ± 165.62 d (486.30–696.76)	495.55 ± 140.98 a (405.98–585.13)	544.86 ± 133.63 c (459.96–629.77)	“a” *p* = 0.018 “b” *p* = 0.012 “c” *p* = 0.010 “d” *p* = 0.033	F(1,11) = 8.650 F(1,11) = 12.543	“a,d” = 0.440 “b,c” = 0.533
MPV 80% 1 RM	0.27 ± 0.08 (0.22–0.32)	0.42 ± 0.09 a (0.36–0.48)	0.34 ± 0.08 (0.29–0.39)	0.43 ± 0.12 c (0.35–0.51)	“a” *p* = 0.001 “c” *p* = 0.009	F(1,11) = 21.850	“a,c” = 0.665
Vmax 80% 1 RM	0.40 ± 0.11 (0.33–0.47)	0.64 ± 0.15 a (0.54–0.73)	0.48 ± 0.12 (0.41–0.56)	0.62 ± 0.17 c (0.51–0.73)	“a” *p* = 0.001 “c” *p* = 0.004	F(1,11) = 24.150	“a,c” = 0.687
Power 80% 1 RM	284.46 ± 101.10 (220.22–348.69)	438.42 ± 96.26 a (377.25–499.69)	355.42 ± 97.65 (293.37–417.46)	452.71 ± 127.34 c (371.80–533.62)	“a” *p* < 0.001 “c” *p* = 0.006	F(1,11) = 29.590	“a,c” = 0.729

Values are mean ± SD and 95% CI of 12 participants; “a”, ”b”, ”c”, “d”: statistically significant differences (*p* < 0.05); η2p = partial eta square (high effect 0.25 to 0.50 and very high effect > 0.50).

## Data Availability

The data that support this study can be obtained from the following address: www.ufs.br/Department of Physical Education, accessed on 12 June 2023.

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
