# Peer review of "Evaluation of the Training Session in Elite Paralympic Powerlifting Athletes Based on Biomechanical and Thermal Indicators"

_sports, 2023, doi:10.3390/sports11080151_

Round 1

Reviewer 1 Report

Organization of the document: The structure of the document seems somewhat disorganized, making it difficult for readers to follow the flow of information. I recommend reorganizing the sections and using clear headings to improve readability and coherence.

Clarity of language: Some sentences and paragraphs appear to be ambiguous or overly complex, which could lead to confusion among readers. I suggest revising the language to ensure clarity and simplicity while conveying the intended message effectively.

Data analysis and interpretation: While the data presented in the report is comprehensive, the analysis and interpretation of the findings could be further strengthened. Consider providing more context and explanations for the data trends and correlations observed, allowing readers to fully understand the implications.

Visual presentation: The visual elements, such as charts and graphs, could be enhanced to better illustrate the data and support the key points being made. Ensure that the visuals are labeled clearly, have appropriate scales, and align with the information presented in the text.

Conclusion and recommendations: The conclusion section should provide a concise summary of the key findings and their implications. Additionally, it would be beneficial to include specific recommendations based on the results, outlining actionable steps for further improvement or decision-making.

Author Response

Dear
Initially I would like to thank you for the review, and all suggestions have been accepted.

Best Regards

Reviewer 2 Report

This study aimed to assess thermal asymmetry and dynamic force parameters in Paralympic powerlifters. Twelve elite male athletes were evaluated before and after a training session using thermography for skin temperature and force dynamics indicators (Average Propulsive Velocity, Maximum Velocity, and Power). Results showed no thermal asymmetry, but differences were observed in force indicators with 45% 1RM, both in terms of asymmetry and time points. With 80% 1RM, asymmetry was observed, but no significant differences between time points. Overall, Paralympic powerlifting was deemed safe in terms of asymmetries.

I have carefully reviewed your study, which focuses on a fascinating and relatively unexplored topic: powerlifting performed by Paralympic athletes. While the subject matter is intriguing, I failed to grasp the innovative aspect and the accompanying rationale. In the introduction, you mention how thermal asymmetry can lead to a decrease in performance. Although this mechanism can be attributed to certain endurance sports and strength training overall, I question its role in maximal performances lasting only a few seconds. This link is entirely missing in the introduction and is not clarified in the Discussion section.

Regarding the thermal aspect, the initiation of the inflammatory process is listed as a mechanism in the Discussion (although the onset times appear to be later than the POST measurements used in the study). Could this simply be the effect of increased blood flow resulting from the recent exertion and the heat generated by muscular work itself?

Another aspect that perplexes me is that all values of muscular performance are higher in the non-dominant limb compared to the dominant limb. Is this observation accurate?

Overall, while the topic of your study is interesting, further clarification and development of the rationale behind thermal asymmetry and its impact on maximal performances would greatly enhance the manuscript. Additionally, addressing the concerns raised above regarding the inflammatory process and the asymmetry of muscular performance would strengthen the overall scientific rigor of your work.

  1. Minors.

  2. Ensure the inclusion of the ANOVA results: It is essential to provide a clear and comprehensive presentation of the statistical analysis. Please include the results of the ANOVA, including F-values, degrees of freedom, and p-values. This will allow readers to understand the overall significance of the observed differences.

  1. Improve the clarity of the result tables

  1. Detailed description of the 1-RM testing protocol: Provide a step-by-step account of the procedure used to determine the 1-RM values. Include information such as the warm-up protocol, the number of attempts allowed, the rest intervals between attempts, and any specific instructions given to the participants during the testing session.

    1.  

  1.  

  2.  

Some sentences are quite confusing. An English grammar revision is required.

Author Response

(The authors gave the same response as above.)

Reviewer 3 Report

In the introduction, I miss general information about muscle strength. Suggests getting acquainted with:
- Zemkova E. Reliability of a new method for assessing muscle strength and velocity during trunk rotation in the sitting position. Physical Activity Review 2019; 7: 1-8. doi: 10.16926/par.2019.07.01
- Studencki M, Ignatjeva A, Nitychoruk M, Golas A, Smolka W, Maszczyk A. Effect of bench press at a specified movement tempo on post-exercise testosterone and cortisol levels. Phys Activ Rev 2021; 9(2): 111-119. doi: 10.16926/par.2021.09.27
- Zemkova E. Science and practice of testing stability and core strength. Physical Activity Review 2018; 6: 181-193. doi: 10.16926/par.2018.06.23

line 182, 195: (Table 2, 3) There is a different symbol in the table (ηp2) and another in the description at the bottom (η2p). Please unify. Data presented in a vague manner. Probably the table contains: data range. Not sure what statistical significance refers to.

Author Response

(The authors gave the same response as above.)
